# The Origin and Regulation of Neuromesodermal Progenitors (NMPs) in Embryos

**DOI:** 10.3390/cells13060549

**Published:** 2024-03-21

**Authors:** Hisato Kondoh, Tatsuya Takemoto

**Affiliations:** 1Biohistory Research Hall, Takatsuki 569-1125, Japan; 2Osaka University, Suita 565-0871, Japan; 3Laboratory for Embryology, Institute for Advanced Medical Sciences, Tokushima University, Tokushima 770-8503, Japan

**Keywords:** neuromesodermal progenitors, NMPs, sinus rhomboidalis, cordoneural hinge, N1 enhancer, Sox2, Bra

## Abstract

Neuromesodermal progenitors (NMPs), serving as the common origin of neural and paraxial mesodermal development in a large part of the trunk, have recently gained significant attention because of their critical importance in the understanding of embryonic organogenesis and the design of in vitro models of organogenesis. However, the nature of NMPs at many essential points remains only vaguely understood or even incorrectly assumed. Here, we discuss the nature of NMPs, focusing on their dynamic migratory behavior during embryogenesis and the mechanisms underlying their neural vs. mesodermal fate choice. The discussion points include the following: (1) How the sinus rhomboidals is organized; the tissue where the neural or mesodermal fate choice of NMPs occurs. (2) NMPs originating from the broad posterior epiblast are associated with *Sox2* N1 enhancer activity. (3) Tbx6-dependent *Sox2* repression occurs during NMP-derived paraxial mesoderm development. (4) The nephric mesenchyme, a component of the intermediate mesoderm, was newly identified as an NMP derivative. (5) The transition of embryonic tissue development from tissue-specific progenitors in the anterior part to that from NMPs occurs at the forelimb bud axial level. (6) The coexpression of Sox2 and Bra in NMPs is conditional and is not a hallmark of NMPs. (7) The ability of the NMP pool to sustain axial embryo growth depends on Wnt3a signaling in the NMP population. Current in vitro models of NMPs are also critically reviewed.

## 1. Introduction

Recent studies have indicated that central nervous system (CNS) tissues and the primary mesoderm components of the embryonic trunk develop from common progenitor cells called neuromesodermal progenitors (NMPs). This discovery caused a revolutionary change in the understanding of cell lineage regulation during embryonic organogenesis. This discovery also impacted the in vitro modeling of embryogenesis starting from pluripotent stem cells. However, there has been a serious misunderstanding regarding the nature and regulation of NMPs, e.g., how the *Sox2* and *Bra* (*T*) genes are involved in NMP regulation, while other critical issues, e.g., how neural vs. mesodermal NMP fate choices are organized at the tissue level, have been left undiscussed. This situation arose primarily because many models concerning NMP regulation are based on snapshot data from E8.5 (2~6 somites) mouse embryos; the period before embryo turning (i.e., through E8.5) is the only time window in mouse embryos when the NMP-containing sinus rhomboidalis tissues are amenable to tissue manipulation. Moreover, at E8.5 of mouse embryogenesis, the NMPs are not yet at the stage of producing neural or mesodermal tissues.

The authors have investigated the regulation of the NMPs for more than two decades, using chicken embryos for continuous observations from the origination of NMPs to a late stage of NMP-derived neural and mesodermal development and mouse embryos for the genetic manipulation of embryogenesis. The NMP studies started with the following series of observations. In 2006, we reported our observation in chicken embryos that *Sox2* N1 enhancer-labeled cells, precursors of the trunk neural tube, also ingress massively into the mesodermal compartment [1]. This report was followed by embryo-wide cell lineage analysis in mouse embryos [2] using a LaacZ transgene [3] inserted into the ROSA26 locus [4], which demonstrated cell clones consisting of both neural and paraxial mesodermal cells at the trunk level, providing definitive evidence for the existence of NMPs (Figure A1). We then reported that *Tbx6*^−/−^ mutant mouse embryos develop extra neural tubes at the expense of the paraxial mesoderm [5] because the N1 enhancer is dysregulated in the NMP-derived mesoderm after gastrulation (failure to turn off the N1 enhancer) [6], indicating that all neural and paraxial mesodermal tissues in the trunk are derived from NMPs.

This review discusses the following issues concerning NMP regulation by comparing chicken and mouse embryo data. (1) The tissue organization of the sinus rhomboidalis is delineated; the neural or mesodermal fate choice of NMPs occurs in the posterior half under the regulation of N1 enhancer activation by Wnt and Fgf signaling and via the repression of *Sox2* activation by BMP signaling. (2) NMPs originating from the broad posterior epiblast are associated with *Sox2* N1 enhancer activity. (3) NMP-derived paraxial mesoderm development in mouse embryos occurs due to Tbx6-dependent *Sox2* repression. (4) A component of the intermediate mesoderm, the nephric mesenchyme, also develops from NMPs. (5) The transition of embryonic tissue development from tissue-specific precursors in the anterior part to that from NMPs occurs around the forelimb bud. (6) The coexpression of Sox2 and Bra in NMPs is conditional and does not provide the hallmark of NMPs; *Sox2* and *Bra* do not reciprocally inhibit each other in NMPs. (7) The ability of NMPs to sustain the axial growth of embryos by their multiplication depends on Wnt3a signaling in the NMP cell population. Current in vitro models of NMPs are also critically discussed.

NMPs play essential roles in generating posterior trunk structures in all vertebrates. However, significant differences in the details of cell regulation have been found between tetrapod animals and fish [7,8]. In this article, we confine our discussion to tetrapod animals.

## 2. The Tissue Organization of the Sinus Rhomboidalis, Where NMPs Undergo a Neural vs. Mesodermal Dichotomous Fate Choice

The sinus rhomboidalis is a tissue complex formed at the growing posterior end of the trunk neural plate anterior to the primitive streak. In this complex, the development of NMPs into cells with a mesodermal or neural fate is regulated, as detailed below. Chicken embryo studies are discussed here, but similar events occur in mouse embryos.

Gastrulation starts with the development of a primitive streak at st. 3 of chicken embryogenesis, where some nonaxial mesoderm and endoderm cells ingress [9,10]. Then, at the onset of st. 4, the node is formed at the anterior end of the streak as a thickened, inverted U-shaped tissue.

### 2.1. The Distinction between the Node and the Chordoneural Hinge (CNH)

The node tissue at st. 4 develops primarily into two secondary tissues, the anterior mesendoderm (AME), which further develops into the prechordal plate (PP) and anterior notochord (ANC), and the posterior notochord (PNC) [11], as determined by replacing the entire node with that of transgenic quail embryos where mCherry is expressed in whole embryonic cells [12] (Figure 1A). The AME extends anteriorly during st. 5 and further develops into the PP underlying the embryonic forebrain and midbrain, and the ANC underlying the hindbrain during st. 8 (Figure 1A). The PNC and the floor plate of the future spinal cord elongate posteriorly starting from st. 6. As the PNC grows, the node tissue is exhausted. At the posterior ends of the PNC and floor plate (FP), these tissues fuse to form the chordoneural hinge (CNH) tissue, which serves as the proliferating progenitor to the notochord and the floor plate [13]. However, in mouse embryos, the dorsal and ventral portions of the chordoneural hinge contribute to FP and PNC development, respectively [14].

During st. 5 and onward, as the primitive streak is shortened toward the posterior side, the CNH position is also displaced posteriorly. The CNH located anterior to the primitive streak is often mistakenly considered the node, but the node tissue is lost as the PNC develops.

### 2.2. Sinus Rhomboidalis Tissue Organization

After the development of several somite pairs, the “sinus rhomboidalis” tissue structure forms around the CNH, which is recognized as a rhomboidal tissue depression. Panels in Figure 1B,C show the dorsal view of a sinus rhomboidalis as seen via scanning microscopy (B) and transverse sections at the respective axial levels (C) reproduced from Catala et al. [13]. A pit (Hensen’s pit) marking the underlying CNH develops at the center of the sinus rhomboidalis (Figure 1B), which is often mistakenly referred to as the “node”. The term “chordoneural hinge” (CNH) is used in this review to avoid confusion.

The change in sinus rhomboidalis tissue organization from the posterior side is shown in Figure 1C. The level (a) is the anterior limit of the primitive streak, showing limited ingression of epiblast cells into the mesodermal layer. Level (b) is between the pit and the streak (referred to as the node-streak border by some researchers [15], where the epiblast and cells ingressing into the mesodermal compartment, both representing NMPs, as discussed in the next section, constitute a large cell mass without making a border (Figure 1C(b), arrowhead). At the CNH (c) level, cell ingression from the epiblast is complete, and the middle part is occupied by the CNH, in which the PNC and FP remain unseparated. Lateral to the CNH, the neural plate and mesodermal tissues start to develop. At level (d), the neural plate and PNC (No) are separate, and the presomitic mesoderm (So) has developed as a solid tissue.

## 3. The Development of Two Distinct Neural Progenitors in the Epiblast with the Activation of the *Sox2* Enhancers N1 or N2, with N1 Marking NMPs

Among approximately 30 distinct neural enhancers regulating *Sox2* [16], enhancers N1 and N2 are activated first in the broad posterior and anterior epiblast region (Figure 2A) [1,16]. The N1 enhancer is activated by the simultaneous action of Wnt and Fgf signaling from neighboring tissues [17,18,19,20] via regulatory elements conserved from mammals to amphibians [1,21] (Figure A2A), while the N2 enhancer is activated by the cobinding of the transcription factors (TFs) Otx2, Pou3/5, and Zic2/3 [16]. The anterior epiblast marked by N2 activity converges to the midline and mostly migrates anteriorly to develop into brain tissue [11] and partly migrates posteriorly to develop into the anterior spinal cord, as indicated by the migration of randomly labeled epiblast cells [11,22,23] (Figure 2B). The posteriorly located N1 enhancer-active cells also migrate mediolaterally (Figure 2B,C) and eventually converge to the sinus rhomboidalis (Figure 2D), and activate *Sox2* at the posterior end of the neural plate [1]. However, the cross-section of the N1 enhancer-labeled chicken embryos at the posterior position of the sinus rhomboidalis showed that a fraction of the mesodermal cells were labeled by the wild-type N1 enhancer (Figure 2E(a), arrowheads). Moreover, using the E-mutant N1 enhancer (with a mutation downstream of the Fgf-responsive element), the large majority of cells in the mesodermal compartment were labeled by N1 enhancer activity (Figure 2E(b), arrowheads) [1].

This finding indicated that (1) the N1 enhancer-labeled cells located in the posterior sinus rhomboidalis have dual potential for both neural and mesodermal development and that (2) wild-type N1 enhancer activity is repressed after ingression into the mesodermal compartment, but the E-mutant N1 enhancer is defective in this repression mechanism. This finding was the first indication of NMP-dependent production of the trunk neural tube and the (paraxial) mesoderm. Given this observation, N1 enhancer activity was considered to mark the NMPs, which was supported by the observation by Brown and Storey [24], where the DiI-labeled cells located far from the midline (yet in the N1-active zone (Figure 2)) of the st. 6 chicken embryo epiblast layer gave rise to both the neural and mesodermal descendant cells (Figure A3A).

## 4. BMP-Dependent *Sox2* Repression Regulates the Neural or Mesodermal Fates of NMPs

Although the N1 enhancer activity covers the sinus rhomboidalis epiblast, strong *Sox2* activation occurs only in the anterior half (Figure 3A(a)) [1,6]. BMP signaling generally represses neural *Sox2* expression, although its mechanism has not yet been clarified. Given that BMP2 is expressed in the primitive streak (Figure 3B(a)) [25], BMP4 and BMP7 are expressed along the lateral sides of the neural plates (Figure 3B(b,c)) [26], whereas the BMP antagonists Chordin (Figure 3B(d)) [26] and Noggin [27] are expressed in the CNH and notochord. A model in which BMP signaling inhibits *Sox2* activation in the N1 enhancer-active region (and, hence, has the potential to activate *Sox2*) in the posterior sinus rhomboidalis was generated; this BMP signaling-dependent inhibition was terminated by the secretion of the BMP antagonists Chordin and Noggin from the CNH and PNC, resulting in the activation of *Sox2* only in the anterior half of the sinus rhomboidalis.

To test this model, we manipulated BMP signaling and examined the change in *Sox2* expression in the sinus rhomboidalis (Figure 3A(b–d)). The expression of constitutively active BMP receptor kinase (CA-Alk6) resulted in the silencing of *Sox2* expression throughout the CNS (Figure 3A(b)). In contrast, the inhibition of BMP signaling by the antagonist Noggin or the dominant-negative BMP receptor (DN-Alk6) elicited *Sox2* expression throughout the sinus rhomboidalis, reflecting the spatial distribution of N1 enhancer activity (Figure 3A(c,d)). Notably, manipulating the BMP signal by these means did not affect the spatial distribution of N1 enhancer activity. These findings confirmed that N1 enhancer activity potentiates *Sox2* expression but that BMP signaling represses *Sox2* expression in the posterior half of the sinus rhomboidalis.

Figure 4 summarizes the sequence of events in the sinus rhomboidalis from posterior to anterior in the order of developmental progression in response to the action of BMP antagonists compared with Figure 1BC in the dorsal view (Figure 4A) and in the transverse section of chicken embryos (Figure 4B). At level (a) through the posterior sinus rhomboidalis, while the N1-active epiblast cells are poised for *Sox2* expression, their *Sox2* expression is suppressed by BMP signaling. BMP-dependent repression of *Sox2* in the posterior sinus rhomboidalis also occurs in mouse embryos, as ectopic expression of Noggin in the sinus rhomboidalis activates *Sox2* expression [6].

At this stage, massive cell ingression from the epiblast layer to the mesodermal compartment occurs, likely via direct ingression but not necessarily through the midline, a mechanism reported by Iimura et al. [28,29]. During this ingression process, N1 enhancer activity is repressed in the mesoderm compartment via an uncharacterized E element-dependent mechanism in chicken embryos but in a Tbx6-dependent fashion in mouse embryos, as discussed in the following section. This difference between chicken and mouse embryos may be related to the fact that Tbx6L, which is not an ortholog of mouse Tbx6, is employed in regulating the paraxial mesoderm of chicken embryos [30].

At level (b) of Figure 4, the CNH secretes BMP antagonists and allows N1 enhancer-active cells to express *Sox2* to initiate neural plate development. Then, at level (c), the neural plate is furnished with the floor plate derived from the CNH, and the presomitic mesoderm develops from the NMP-derived mesodermal cells. Thus, NMPs do not develop into neural and mesodermal tissues simultaneously but develop primarily into mesoderm in the posterior sinus rhomboidalis, where BMP signaling inhibits neural development, and mainly into neural tissue once *Sox2* expression is activated in the more anterior part of sinus rhomboidalis.

## 5. Tbx6 Represses N1 Enhancer Activity in the NMP-Derived Mesoderm and Inhibits Neural Development of the Paraxial Mesoderm

As shown in Figure 2E for the chicken embryos, the N1 enhancer activity of NMPs is repressed after ingression into the mesodermal component; otherwise, ectopic neural tissues would develop in the mesodermal compartment. The ectopic neural tissues, in fact, develop in *Tbx6*-defective (*Tbx6*^−/−^) mouse embryos due to the failure of N1 enhancer repression in the mesoderm.

Chapman and Papaioannou [5] reported that *Tbx6*^−/−^ mouse embryos produce bilateral extra neural tubes at the expense of the paraxial mesoderm. We investigated N1 enhancer activity in *Tbx6*^−/−^ embryos, and found that N1 enhancer activity was maintained in the mutant mesoderm [6].

Figure 5 shows a comparison of the N1 enhancer activity indicated by N1-*tkEgfp* transgene expression and endogenous *Sox2* expression in E8.5 and E9.5 mouse embryos using in situ hybridization of transcripts [6]. In wild-type embryos, N1 enhancer activity was turned off in the paraxial mesoderm at E8.5, no *Sox2* expression was detected in the paraxial mesoderm at E8.5, and only a normal neural tube developed on the midline at E9.5 (Figure 5a–c). In contrast, the *Tbx6*^−/−^ embryos exhibited N1 enhancer activity in the paraxial mesoderm compartment and *Sox2* expression therein at E8.5, which resulted in bilateral ectopic neural tube development at E9.5 (Figure 5d–f).

To confirm that the ectopic neural tubes developed as a consequence of persistent N1 enhancer activity in the mesodermal compartment in the *Tbx6*^−/−^ embryos, we deleted the N1 enhancer (ΔN1/ΔN1) of the *Tbx6*^−/−^ embryos, which eliminated *Sox2* expression in the paraxial mesoderm at E8.5 and ectopic neural tube development at E9.5 (Figure 5g,h). Our observations indicated that the N1 enhancer must be shut off after NMP ingression into the mesodermal compartment for paraxial mesoderm development to occur and that Tbx6 plays a role in N1 enhancer repression in mouse embryos [6].

The transcription factor (TF) Tbx6 does not bind directly to the N1 enhancer sequence; instead, Tbx6 represses *Wnt3a* expression in the mesodermal compartment, leaving the epiblast as the only cell layer expressing *Wnt3a*. In *Tbx6*^−/−^ embryos, *Wnt3a* is also expressed at a high level in the mesodermal compartment, thus sustaining N1 enhancer activity from the epiblast to the mesodermal compartment [6].

The ΔN1/ΔN1 condition in the *Tbx6*^+/+^ embryos removed *Sox2* expression in the sinus rhomboidalis at E8.5 [31] (as discussed in Section 8), with the *Sox2*-expressing anterior neural tube likely formed from N2-active neural-specific progenitors (Figure 2D). However, *Sox2* expression was activated in the forming neural tube at E9.5, and mice developed normally and were fertile, presumably owing to the combined activity of other *Sox2* neural enhancers activated later than N1 [16].

## 6. Nephric Mesenchyme Also Develops from NMPs

The intermediate mesoderm is a narrow tissue formed between the paraxial and lateral plate mesoderm, which develops into the urogenital system [32]. Histological analysis of *Tbx6*^−/−^ mutant embryos at the posterior trunk level indicated that besides the ectopic neural tubes replacing the paraxial mesoderm, additional Sox2-expressing tubular tissues arise in the region ventrolateral to the paraxial region where the Wolffian duct and surrounding mesonephric mesenchyme normally reside (Figure 6A) [33]. The Wolffian duct primordium expressing Pax2 arises as an intermediate mesoderm immediately posterior to the 7th somite level and extends posteriorly to form the Wolffian duct, guided by the mesonephric mesenchyme expressing WT1 [34,35]. Figure 6 shows that the nephric mesenchyme is also derived from NMP-derived Tbx6-expressing cells (paraxial mesoderm/nephric mesenchyme-common progenitors).

Hayashi et al. [33] showed that paraxial mesoderm-forming Tbx6+ cells express *Foxc2*, whereas nephric mesenchyme-forming Tbx6+ cells express *Osr1* under the influence of lateral plate-derived BMP signaling. Furthermore, the authors demonstrated that *Osr1*^−/−^ embryos expanded the paraxial mesoderm to the mesonephric mesenchyme region and failed to develop WT1-expressing nephric mesenchyme in the trunk region. In contrast, the nephric mesenchyme that developed in *Foxc1*^−/−^; *Foxc2*^−/−^ mutant embryos was expanded into the somitic mesoderm region [36]. In brief, the nephric mesenchyme, a component of the intermediate mesoderm, develops from NMPs via two repression steps, i.e., Tbx6-dependent *Sox2* repression to produce common progenitors of the paraxial mesoderm/nephric mesenchyme and Osr1-dependent *Foxc2* (possibly also *Foxc1*) repression to promote nephric mesenchyme development (Figure 6B). In *Tbx6*^−/−^ embryos, *Pax2*-expressing Wolffian ducts fail to extend to the trunk due to the lack of nephric mesenchyme (Figure 6C).

## 7. The Transition of Spinal Cord and Paraxial Mesoderm Sources from Tissue-Specific Progenitors to NMPs

As shown in Figure 2D, the spinal cord at the anterior end was fully occupied by the cells labeled with N2 enhancer activity (mRFP1), indicating their origin in the brain-forming region followed by posterior migration [22]. However, toward the posterior side, N1 enhancer-labeled (EGFP) cells exhibited an increased contribution to the spinal cord and occupied the st. 8 sinus rhomboidalis of chicken embryos. This finding indicated that the transition of spinal cord precursors occurs from non-NMP- to NMP-derived cells during the posterior extension of the embryo axis. A neural cell lineage tracing study using mouse embryos [37] indicated the existence of a group of neural cell clones dedicated to the spinal cord with anterior limits varying from the cervical to the forelimb bud levels, suggesting that the contributions of the NMP-derived cells to the spinal cord gradually increased and prevailed in the spinal cord posterior to the forelimb level (Figure A3B). The forelimb bud level is the anterior limit of ectopic neural tube development in *Tbx6*^−/−^ embryos (Figure 7B) [5,6], indicating that NMP-dependent neurogenesis, to a large extent, replaces non-NMP-dependent neurogenesis around the forelimb bud.

As discussed below, NMPs presumably multiply in the posterior sinus rhomboidalis in a manner dependent on the activity of Bra and Wnt3a. As schematized in Figure 7C,D, *Bra*^−/−^ and *Wnt3a*^−/−^ embryos develop similar phenotypes where (1) somitogenesis is arrested at approximately the forelimb bud level due to the lack of the paraxial mesoderm at more posterior levels [38,39,40,41,42] and (2) ectopic neural cell masses develop immediately posterior to the last somite pair [41,42]. The first phenotype is accounted for by the anterior contribution of mesoderm-specific precursors that arise from E7.5 to E8.0 via primitive streak-associated gastrulation [31,43], while the second phenotype is explained by neural tissue development from the growth-arrested NMP pool in the *Bra*^−/−^ and *Wnt3a*^−/−^ embryos. The inferior spinal cord develops posterior to the forelimb bud in *Bra*^−/−^ and *Wnt3a*^−/−^ embryos (Figure 7C,D), likely reflecting the contribution of the non-NMP neural precursors initially marked by N2 enhancer activity (Figure 2D) [22]. Taken together, these observations show that the anterior-to-posterior-directed transition from non-NMP- to NMP-derived precursors in the spinal cord and paraxial mesoderm histogenesis occurs at approximately the forelimb level of mouse embryos.

## 8. *Sox2* and *Bra* Coexpression Is Not a Hallmark of NMPs

In chicken embryos, the posterior sinus rhomboidalis coexpresses low levels of *Sox2* and *Bra* [44]. Using E8.5 (2~6 somites) mouse embryos, Wymeersch et al. [45] investigated the distribution of cell populations that develop into both neural and paraxial mesoderm in two days. Their strategy was to excise various tissue samples from the sinus rhomboidalis of E8.5 embryos expressing EGFP to graft them homotopically to the host E8.5 embryo under culture, and to determine whether the grafted tissue developed into both neural and paraxial mesoderm at E10.5. This mapping study identified the area of the sinus rhomboidalis outlined by the broken line in Figure 8A as the tissue site giving rise to both the neural and paraxial tissues.

In this area, Sox2 and Bra were coexpressed at low levels (Figure 8A), but Sox2 expression was greater at more anterior positions after BMP-dependent inhibition was alleviated (Figure 3), and the change in the Bra expression level was reversed. Notably, not all the sinus rhomboidalis tissues coexpressing Sox2 and Bra produced neural and paraxial mesoderm tissues in the assay. A comparison of *Sox2* and *Bra* expression, as observed through in situ hybridization, with N1 enhancer activity (Figure 8B(a–c)), confirmed the above expression profiles, and the same relationship between N1 enhancer activity and Sox2 expression was observed as in chicken embryos (Figure 3). On the other hand, Rodrigo Albors et al. [46] traced the descendants of *Nkx1-2*-expressing cells that cover the sinus rhomboidalis posterior epiblast. The reporter-labeled cells expressing Nkx1-2 around E8 were distributed widely in the spinal cord and mesodermal tissues of E9.5 embryos, confirming the neuromesodermal bipotentiality of the posterior sinus rhomboidalis.

As discussed in Section 7, the contribution of NMPs to the neural and mesodermal tissues becomes substantial posterior to the forelimb (12~15 somites) level. Therefore, it is unlikely that the tissue organization in the E8.5 sinus rhomboidalis represents that of the later-stage sinus rhomboidalis, where the NMPs multiply, simultaneously giving rise to the neural and mesoderm tissues. With these reservations, the dual-potential precursors for the neural and mesodermal tissues were located in the Sox2 and Bra coexpression region in E8.5 mouse embryos (Figure 8A).

Guillot et al. [47] performed cell lineage and transcriptome studies by labeling the anterior region of the st. 5 primitive streak-abutting epiblast of chicken embryos where Sox2 and Bra are coexpressed, assuming that they are the NMP pools. However, as discussed above, Sox2-Bra coexpression is not an absolute criterion for identifying NMPs; the epiblast region studied by Guillott et al. [47] included only a small fraction of N1 enhancer-labeled NMPs (Figure 2A). Despite these reservations, the study confirmed the developmental dual potentiality of the majority of labeled cells, which contributed to the neural and mesodermal tissues at posterior trunk levels, reflecting the occupation of the anterior trunk tissues derived from non-NMP precursors, as discussed in Section 7.

The parallelism between NMP localization and the *Sox2* and *Bra* coexpression regions was, thus, highlighted. Consequently, *Sox2* and *Bra* coexpression has been frequently suggested as the hallmark of NMPs, creating confusion. Even a model in which *Sox2* and *Bra* repress each other has been proposed; this cross-antagonism between two TF genes sustains the stemness of NMPs [48,49]. However, the model has been refuted by the following experimental results. (1) *Sox2* expression in the entire sinus rhomboidalis was lost by N1 enhancer deficiency (∆N1/∆N1) (Figure 8B(b,d)), but Bra expression was not affected (Figure 8B(c,e)) [31]. (2) Bra-deficient embryos fail to maintain NMPs owing to a secondary defect in *Wnt3a* activation (see the next section). However, in chimeric embryos with wild-type cells, NMPs can be propagated during embryogenesis owing to Wnt signals from the wild-type cells, although the *Bra*^−/−^ NMPs produce neural tissues but not paraxial mesoderm. In these chimeric embryos, *Bra*^−/−^ NMPs did not increase *Sox2* expression above the wild-type level [50].

As discussed in Section 4, whether NMPs express *Sox2* depends on whether the BMP signals suppressing *Sox2* act on NMPs. BMP-dependent regulation appears to differ in the tail bud, which develops after neuropore closure (in chicken embryos after st. 13/14 [51]). Kawachi et al. [52] demonstrated that in the chicken tail bud, Sox2+ NMP populations were the immediate precursors for neural and mesodermal tissues, whereas the Sox2+ population gave rise to mesodermal tissues. These findings indicate that the coexpression of Sox2 and Bra in NMPs is conditional and cannot be considered a hallmark of NMPs. Cell labeling based on N1 enhancer activity in NMPs will provide a more reliable marker for NMPs.

## 9. NMP Pool Maintenance during Axial Elongation of the Embryo through the Supply of Canonical Wnt Signals

Pioneering studies by Wilson and Storey groups [15,44,53] indicated that the cells in the posterior sinus rhomboidalis (“node-streak border” in the original papers), namely, NMPs, multiply and contribute to embryo axial elongation. The question is how the proliferation of NMPs is regulated.

Axial elongation in both *Wnt3a*^−/−^ and *Bra*^−/−^ mutant embryos is arrested where posterior trunk elongation starts to change from NMP-independent to NMP-dependent developmental processes, as discussed in Section 7 [40,42] (Figure 7). These observations indicate that the mechanism by which NMP cell multiplication leads to cell pool expansion depends on the *Wnt3a*↔*Bra* coregulatory loop that is formed in axial progenitor (NMP) tissue [42] (Figure 9A). The pools of NMPs in the sinus rhomboidalis of *Wnt3a*^−/−^ or *Bra*^−/−^ mutant embryos do not expand but develop into a mass of neural tissue, with mesodermal development being blocked in the absence of the Bra TF (Figure 7).

A critical question is how the *Wnt3a*↔*Bra* coregulatory loop participates in NMP pool expansion. According to an analysis of *Bra*^−/−^ (tdTomato-labeled)↔wild-type chimeric embryos, *Bra*^−/−^ cells proliferate and contribute to the spinal cord and posterior NMP pools (although *Bra*^−/−^ NMPs are defective in mesoderm production) [50]. These findings indicate that cell exogenous Wnt signaling is sufficient for NMP cell proliferation and that Bra functions to activate *Wnt3a* expression.

Moreover, intercellular exchange of Wnt signals is required for stable propagation of the NMP pool. In a recent study, Hatakeyama et al. [54] produced a knock-in mouse line in which the Wnt3a-Fzd5 fusion sequence was used to replace the Wnt3a gene sequence. The Wnt3a-Fzd5 fusion protein is not secreted, but it mimics the Wnt3a-bound receptor Fzd5; hence, exclusively cell autonomous Wnt signaling was established in mice. Under homozygous Wnt3a-Fzd5 knock-in conditions, Wnt signaling in individual NMP cells became highly heterogeneous, as measured by the WntVis system [55]. These cells produced many Wnt signal-low cells in which Wnt3a↔Bra coregulation possibly entered the negative loop. Consequently, NMP maintenance became impaired posterior to the hindlimb, although some NMPs inferior to the wild-type NMP pool were maintained at the tip of the tail bud up to E13.5. These findings indicated that mutual Wnt signaling among NMP cells results in the maintenance of high Wnt signaling in the entire cell population and guarantees the stable maintenance of the NMP pool until embryonic axial elongation is complete (Figure 9B). Interestingly, conditional knockout of *Sall4* using T(Bra)-Cre resulted in phenotypes analogous to those of Wnt3a-Fzd5 knock-in embryos [56,57], suggesting the possible involvement of Sall4-dependent regulation in the Wnt3a secretion.

## 10. In Vitro Models of NMP Development

It is imperative to develop in vitro models of NMPs starting from appropriate pluripotent stem cells to characterize and follow the developmental changes in the gene regulatory networks of NMP cells in detail. In vitro studies are also crucial for developing human NMP models.

In vitro NMP studies using mouse pluripotent stem cells have generally taken approaches to follow Sox2-Bra-coexpressing cells, as described in Section 8 above. Edri et al. [58] compared single-cell transcriptomes of Sox2-Bra-coexpressing cell populations derived from mouse ESCs and EpiSCs via different protocols [59,60,61] with those of the E8.5 posterior sinus rhomboidalis. The transcriptome profiles were highly divergent within and among the Sox2-Bra-coexpressing cell populations, in contrast to the tight clustering of the E8.5 posterior sinus rhomboidalis cells, which showed limited overlap with Sox2-Bra-coexpressing cells in the transcriptome profile, indicating the limitation of following Sox2-Bra-coexpressing cells. Notably, the *Cdx* family and *Nkx1-2* TF genes are commonly expressed in Sox2-Bra-coexpressing cell populations [46,48,49,58].

Human NMP-like cells produced in vitro are currently used pragmatically to produce neural or mesodermal tissues with defined anteroposterior characteristics [62,63,64,65,66,67]. Notably, the empirically determined culture conditions match the N1 enhancer and *Sox2* activation in the NMPs and the downregulation of N1 and *Sox2* in the nonneural tissues. A representative protocol for initiating in vitro NMP-like cell development from human ES/iPS cells [63] includes the inhibition of Tgfβ signaling (inhibition of nodal signaling) and BMP signaling (to activate *Sox2*, as discussed in Section 4; human ES/iPS cells have endogenous BMP signaling) in the presence of Fgf2, which promotes neural development [61], and the production of a strong Wnt signal through the addition of CHIR99021 (eliciting N1 enhancer activation in combination with Fgf2 [1,6]). NMP-like cells expressing Sox2 and Bra appear transiently in culture but cannot be maintained. By shifting the culture conditions of transient NMP-like cells to those of developmentally more advanced cell groups, neural tissues are generated with the addition of retinoic acid, and mesodermal tissues are generated by inhibiting Fgf signaling (the condition for inactivating N1 enhancer activity). The axial characteristics of the NMP-derived tissues reflected by the *Hox* gene expression profiles are regulated by the Wnt (modulating anteroposterior characteristics) and Gdf11 (activating posterior characters) signaling inputs into the cells in culture. A successful example of applying this strategy in generating neural tissues with discrete anteroposterior and dorsoventral regional specificity was found in the study by Iyer et al. [67]. Using the above protocols, the cervical neural tissues derived from neural-specific progenitors [22,43] were not produced (see Section 7).

## 11. Conclusions

In this article, we presented a curated collection of published results on NMPs, providing proper context for readers to have a unified view of NMPs. Several points are emphasized. (1) NMPs in the posterior sinus rhomboidalis produce mesodermal tissues at the posterior side, where *Sox2* expression is inhibited by BMP signaling, and neural tissues at more anterior sites, where BMP-dependent *Sox2* repression is terminated. (2) The association of Sox2 and Bra coexpression with NMPs is conditional and is not a hallmark of NMPs. Cell labeling based on the specificity of the *Sox2* N1 enhancer will be useful for identifying NMPs more reliably. (3) NMPs are derived from a broad area of the posterior epiblast and are not given birth in the sinus rhomboidalis tissue. (4) NMPs serve as neural and mesodermal tissue precursors posterior to the forelimb level after anterior tissues are developed from tissue-specific precursors.

We hope that the information in this article will be helpful to a wide range of scientists interested in and involved in NMP research.

## Figures and Tables

**Figure 1 cells-13-00549-f001:**
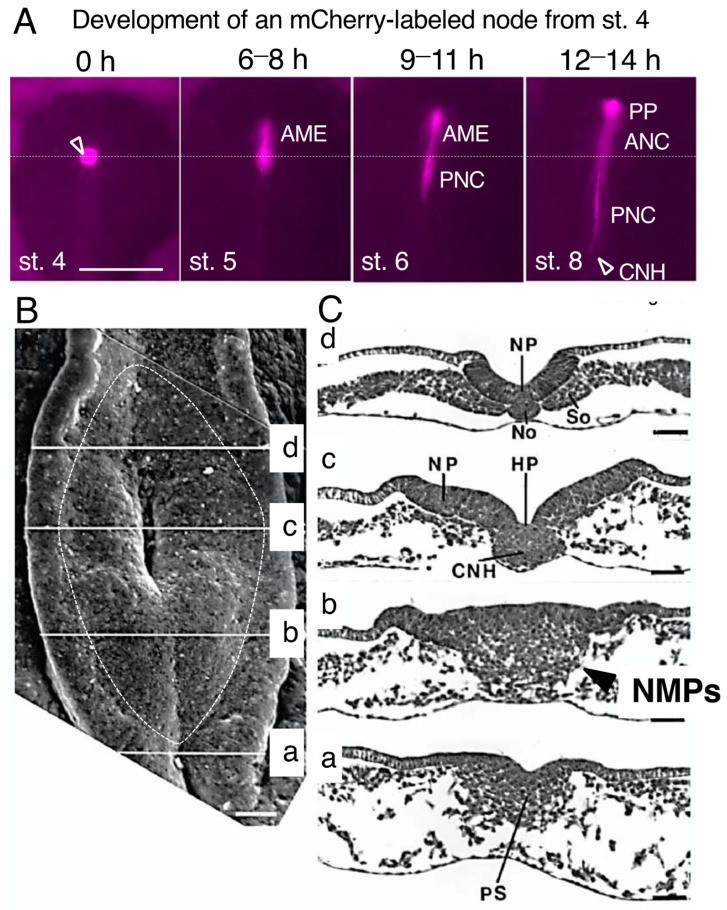
The tissue organization of the sinus rhomboidalis, with the chordoneural hinge (CNH) at the center. (**A**) The entire node tissue at. st. 4 was labeled by exchanging the node with that of stage-matched transgenic quail expressing mCherry in all embryonic cells [12]. The anterior mesendoderm (AME) developed from the node at early st. 5 and later gave rise to the prechordal plate (PP) and anterior notochord (ANC), whereas the posterior notochord (PNC) developed from the node during st. 6. With the development of the PNC, the node tissue was lost. The CNH was located at the posterior end of the PNC. The horizontal dotted line indicates the axial level of the original node. The bar represents 500 µm. Data from [11]. (**B**) Dorsal view of the sinus rhomboidalis as seen via scanning microscopy. The sinus rhomboidalis is formed at the growing posterior end of the spinal cord with the CNH at the center at level (c). The data are from Figure 1 of [13] and were reproduced with permission from Development. The bar represents 50 µm. (**C**) Transverse histological sections through the sinus rhomboidalis at different axial levels in (**B**). NP, neural plate; No, posterior notochord; So, presomitic mesoderm; HP, Hensen’s pit; CNH, chordoneural hinge; PS, primitive streak. These tissue labels are from the original publication [13]. The bar represents 50 µm. The arrow in (b) indicates the mass of cells that ingressed around the primitive streak position into the mesodermal compartment, which represents NMPs.

**Figure 2 cells-13-00549-f002:**
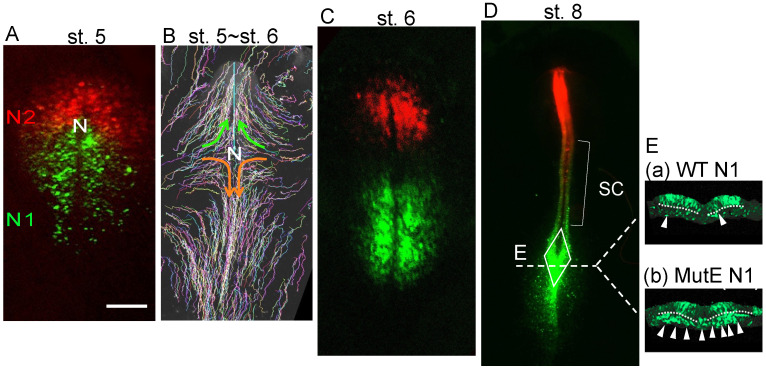
The distribution and migration of N1 enhancer-labeled epiblast cells and N2 enhancer-labeled epiblast cells, which represent precursors of NMPs and neural-specific progenitors. (**A**,**C**,**D**) Chicken embryos were coelectroporated with N1-tkEGFP and N2-tkmRFP1 vectors at st. 4 [16], and fluorescence images were recorded at different developmental stages (original data). (**B**) Epiblast cells were randomly and sparsely labeled with a Supernova vector cocktail [11,23], and the cell trajectories from st. 5 to st. 6 are displayed. The data are taken from Nakamura et al. [22]. N, node position. The bar represents 500 µm. (**A**) N2 enhancer-active cells cover the anterior epiblast, whereas N1 enhancer-active cells occupy a broad region of the posterior epiblast. (**B**) Brain-forming cells, all derived from N2-expressing epiblasts, migrated medioanteriorly (green arrows), whereas spinal cord-forming cells (a subset of N2-labeled cells and the majority of N1-labeled cells) migrated medioposteriorly (orange arrows). (**C**) The distribution of cells labeled by N1 and N2 enhancer activity at the beginning of neural plate formation at st. 6. (**D**) At st. 8, the neural tube was formed, in which the spinal cord (SC) was distributed by both N2-labeled cells showing anterior enrichment and N1-labeled cells showing posterior enrichment. At the posterior growing end of the spinal cord, the sinus rhomboidalis, indicated by the rhombus, was formed where most N1-active cells converged. Line E indicates the axial level of the transverse sections shown in (E). (**E**) Transverse sections of the posterior sinus rhomboidalis labeled with the wild-type N1 enhancer (**a**) or an E-mutant N1 enhancer (**b**) [1]. The broken lines indicate the position of the basement membrane separating the upper epiblast layer and the mesodermal compartment. The cells were wild-type, and, hence, the difference in the fluorescence distribution reflects the difference in the regulation of enhancer activity. (**a**) Wild-type N1 enhancer activity was observed in most of the upper layer cells of the sinus rhomboidalis and in a fraction of cells in the mesodermal compartment (white arrowheads), indicating that N1 enhancer-active, and, hence, *Sox2* expression-potentiated, cells contribute to mesoderm development. (**b**) Using the E-mutant N1 enhancer, the majority of cells in the mesodermal compartment showed N1 enhancer activity (white arrowheads), indicating that *Sox2* expression was potentiated in virtually all mesodermal compartment cells among the sinus rhomboidalis cells and that the activity of the N1 enhancer was repressed in the mesodermal compartment.

**Figure 3 cells-13-00549-f003:**
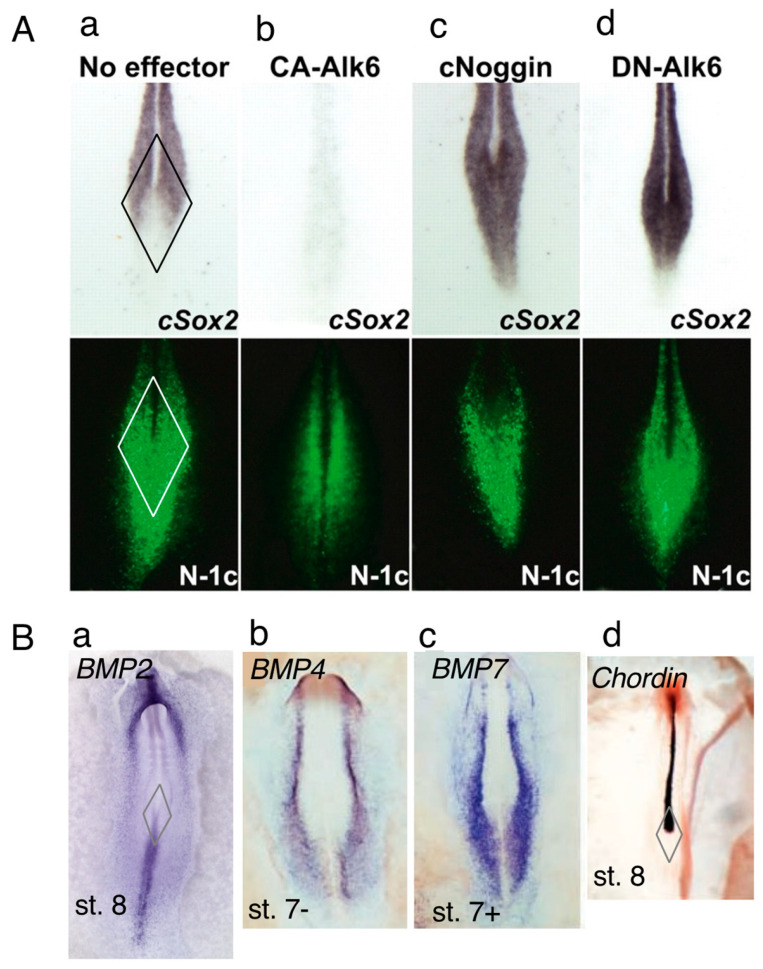
*Sox2* expression is repressed by BMP signaling in the posterior sinus rhomboidalis. (**A**) *Sox2* expression (upper, in situ hybridization) and N1 enhancer activity (lower, EGFP expression using the trimeric N1 core (N1-c) enhancer) in the sinus rhomboidalis of st. 8 chicken embryos. (**a**) With no exogenous effector, *Sox2* expression in the posterior sinus rhomboidalis was reduced. (**b**) The expression of constitutively active Alk6, which mimics strong BMP signaling, after st. 5 eliminated *Sox2* expression in the entire embryo, demonstrating strong inhibition of neural *Sox2* expression by BMP signaling. (**c**,**d**) The expression of the BMP antagonist Noggin or dominant-negative Alk6 after st. 5 elicited a high level of *Sox2* expression in the posterior sinus rhomboidalis, indicating that the reduction in *Sox2* expression resulted from repression by BMP signaling. Figure panels were reproduced from [1]. (**B**) Potential sources of BMP and BMP antagonists regulating *Sox2* expression in the sinus rhomboidalis. The formation of the sinus rhomboidalis is indicated by the rhombus in (**a**,**d**). (**a**–**c**) The expression profiles of *Bmp2* [25], *Bmp4* [26], and *Bmp7* [26] in st. 7 to st. 8 chicken embryos, among which the *Bmp2* expressed in the posterior sinus rhomboidalis and primitive streak is a strong candidate for the *Sox2*-repressing BMP in the posterior sinus rhomboidalis. (**d**) Chordin [26] and Noggin [27], not shown here, are expressed in the notochord to the CNH, presumably relieving *Sox2* from BMP-dependent inhibition. (**a**) from the GEISHA database [25], and (**b**–**d**) are adapted from [26] in Development with permission.

**Figure 4 cells-13-00549-f004:**
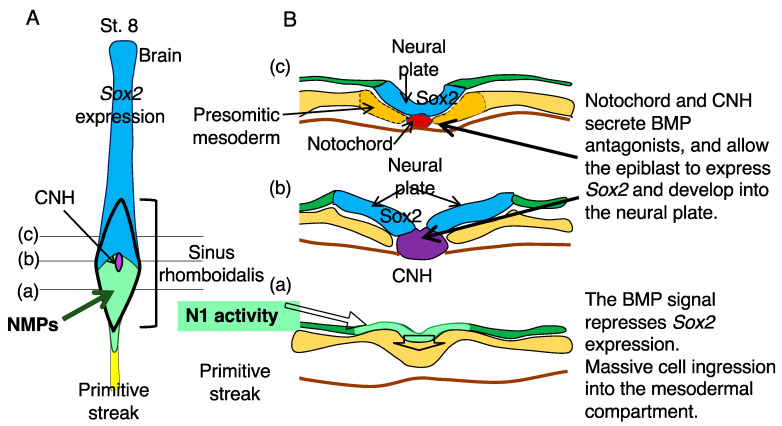
The sequence of cellular events that occur posterior to anterior in the sinus rhomboidalis. (**A**) A schematic of the developing CNS of a chicken embryo at st. 8, where the sinus rhomboidalis and primitive streak positions are indicated. (**B**) Diagrams of transverse sections at levels (**a**–**c**) in (**A**) from posterior to anterior. (**a**) The BMP signal represses *Sox2* expression at this level, although epiblast cells exhibit N1 enhancer activity (light green). This level is where the massive ingression of cells into the mesodermal compartment occurs. (**b**) At this level, the CNH secretes the BMP antagonists Chordin and Noggin, terminating BMP-dependent *Sox2* repression and allowing N1 enhancer-active cells to develop into neural plates (blue). (**c**) At this level, the CNH developed into the posterior notochord (red) and the floor plate, the latter fusing with the bilateral neural plate and forming a continuous neural plate.

**Figure 5 cells-13-00549-f005:**
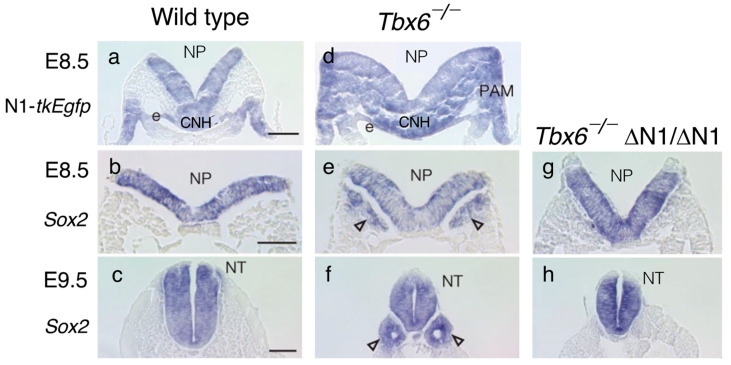
N1 enhancer activity and *Sox2*-expressing neural tube development in wild-type and *Tbx6*^−/−^ embryos shown in trunk cross sections. (**a**,**d**) The detection of N1 enhancer activity by N1-*tkEgfp* transcripts via in situ hybridization. (**a**) No N1 enhancer activity was detected in the wild-type paraxial mesodermal compartment. (**d**) In contrast, N1 activity was widespread in the *Tbx6*^−/−^ mesodermal component. (**b**,**e**,**g**) *Sox2* expression in E8.5 embryos. (**b**) *Sox2* was expressed only in the neural plates (NPs) of wild-type embryos. (**e**) However, in *Tbx6*^−/−^ embryos, the bilateral tissues of the paraxial mesoderm positions also expressed *Sox2* (arrowheads). (**g**) Ectopic *Sox2* expression was lost by introducing ΔN1/ΔN1 (N1 enhancer deletion) to *Tbx6*^−/−^ embryos. (**c**,**f**,**h**) The development of *Sox2*-expressing neural tubes in E9.5 embryos reflected the *Sox2* expression profiles in E8.5 embryos. (**c**) Wild-type embryos had a single neural tube (NT) (**f**). The *Tbx6*^−/−^ embryos developed a central neural tube and two additional neural tubes (arrowheads). (**h**) *Tbx6*^−/−^; ΔN1/ΔN1 embryos developed only a single neural tube. The data were adapted from [6] with permission from Springer Nature.

**Figure 6 cells-13-00549-f006:**
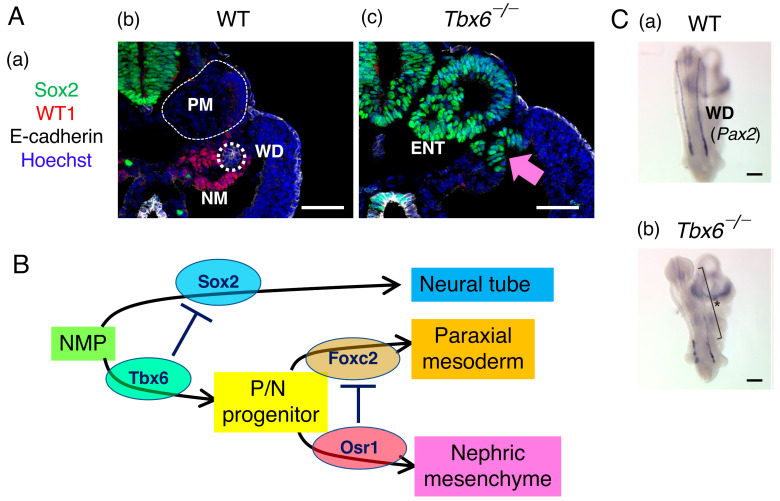
Nephric mesenchyme, a population of intermediate mesodermal cells, develops from the NMPs. (**A**) The impact of a defect in Tbx6 expression (*Tbx6*^−/−^) on intermediate mesoderm development. (**a**) The markers used in the histological analysis. Sox2, neural tissue; WT1, nephric mesenchyme; E-cadherin, the Wolffian duct and the gut epithelium; Hoechst, cell nuclei. (**b**,**c**) Histological cross sections at the posterior trunk level of wild-type (**b**) and *Tbx6*^−/−^ (**c**) embryos stained for the markers indicated in (**a**). PM, paraxial mesoderm; WD, Wolffian duct (white dotted circle); NM, nephric mesenchyme; ENT, ectopic neural tube. In *Tbx6*^−/−^ embryos, the paraxial mesoderm and nephric mesenchyme tissues present in wild-type embryos were replaced by ectopic neural tubes and the Wolffian duct was absent. The pink arrow indicates the ectopic neural tube replacing the nephric mesenchyme. The bars indicate 50 µm. The data were adapted from [33]. (**B**) Schematic summary of the derivation of the neural tube (Sox2+), paraxial mesoderm (Foxc2+), and nephric mesenchyme (Osr1+) via the repressive interactions indicated by the ⏉ marks. (**C**) Wolffian duct development in wild-type and *Tbx6*^−/−^ embryos detected by *Pax2* in situ hybridization. In the *Tbx6*^−/−^ embryo lacking the nephric mesenchyme, the Wolffian duct failed to extend into the posterior region (*). The bars indicate 50 µm. The data were adapted from [6] with permission from Springer Nature.

**Figure 7 cells-13-00549-f007:**
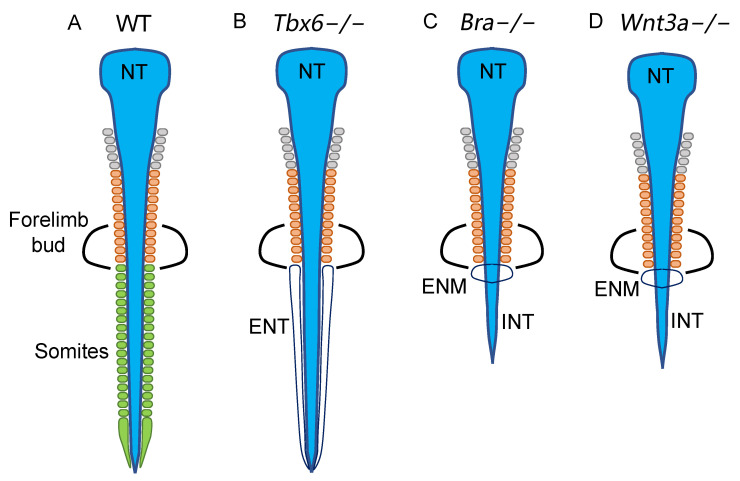
Schematics of the neural and somitic phenotypes of *Tbx6*^−/−^, *Bra*^−/−^, and *Wnt3a*^−/−^ mutants, indicating that the transition of histogenesis occurs from specific progenitor-derived to NMP-derived cells at the forelimb axial level. (**A**) Wild-type embryo, (**B**) *Tbx6*^−/−^ embryo, (**C**) *Bra*^−/−^, and (**D**) *Wnt3a*^−/−^ mutant embryos. Somites of different anteroposterior levels are color-coded gray for the cranial, orange for the cervical, and green for the thoracic somites. NT, neural tube; ENT, ectopic neural tube; INT, inferior neural tube; ENM, ectopic neural cell mass.

**Figure 8 cells-13-00549-f008:**
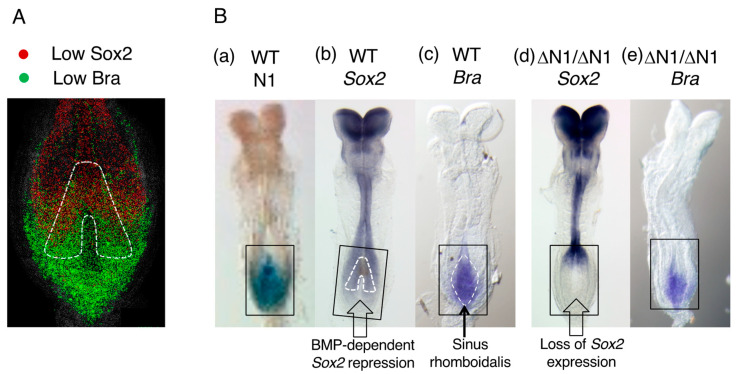
Sox2 and Bra expression in mouse E8.5 NMPs. (**A**) The region of overlap of Sox2 and Bra protein expression in the sinus rhomboidalis of an E8.5 mouse embryo in an optical section. (The expression levels of Sox2 and Bra were assessed as “low” among the expression ranks in the embryos in the original publication, but substantial amounts of these proteins accumulated in the nuclei.) The inverted V region outlined by the broken line indicates where the homotopic grafts produced both neural and paraxial mesoderm tissues in E10.5 host embryos. The figure panel was adapted from Figure 5A(a) of [45]. (**B**) The effect of the loss of N1 enhancer activity (ΔN1/ΔN1) on *Sox2* and *Bra* expression. Black boxes indicate the area of interest. (**a**–**c**) A comparison of N1 enhancer activity, *Sox2* expression, and *Bra* expression detected by in situ hybridization in E8.5 wild-type mouse embryos. In (**b**)**,** the “inverted V” region and, in (**c**)**,** the sinus rhomboidalis are indicated by broken lines. (**d**,**e**) *Sox2* and *Bra* expression in ΔN1/ΔN1 embryos. *Sox2* expression was completely lost in the sinus rhomboidalis in (**d**); however, *Bra* expression in the sinus rhomboidalis was not affected by the loss of *Sox2* expression (**e**). Adapted from Figure 3.11 of [31].

**Figure 9 cells-13-00549-f009:**
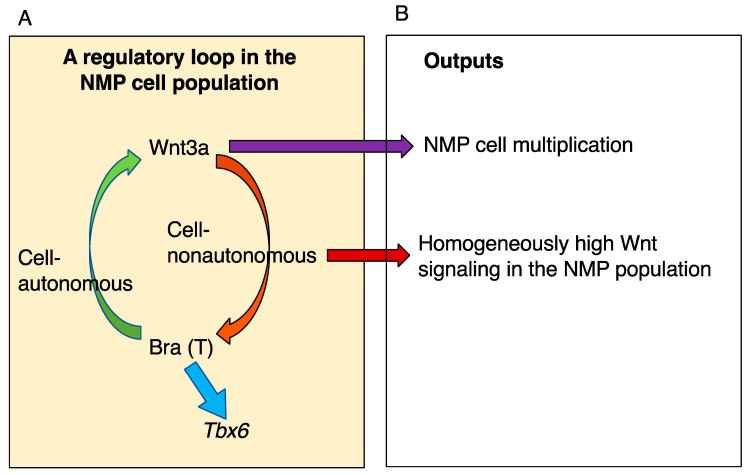
Wnt signaling plays a role in the proliferation of NMPs to sustain the cell population while producing neural and mesodermal cells. (**A**) The Wnt3a-Brachyury coactivating regulatory loop in the NMP (classically axial stem) cell population [42]. (**B**) The exchange of Wnt signals among cells sustains the proliferative NMP cell population. As *Bra*^−/−^ NMPs proliferate and are maintained in chimeric mouse embryos, the primary role of Bra is to activate *Wnt3a* expression, and Wnt3a is a direct requirement for NMP proliferation and maintenance [50]. When intercellular Wnt signal exchange is inhibited, a significant fraction of NMP cells enter the reduced reproductive cycle through attenuated cell autonomous Wnt signaling, and the number of reproducing cells is minimized [54]. Therefore, the intercellular exchange of Wnt signals guarantees homogeneous enrichment of Wnt signaling in the NMP cell population.

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
