# Peer review of "The Origin and Regulation of Neuromesodermal Progenitors (NMPs) in Embryos"

_cells, 2024, doi:10.3390/cells13060549_

Round 1

Reviewer 1 Report

Comments and Suggestions for Authors

This comprehensive review by Kondoh and Takemoto provides a detailed examination of the research findings concerning the origin and regulatory mechanisms governing neuromesodermal progenitors (NMPs) in embryos. The article offers valuable insights into the biology of NMPs, catering specifically to the interests of specialized scientists engaged in NMP research. The clarity of expression and the inclusion of well-crafted original research figures greatly contribute to the elucidation of the intricate processes involved.

However, before publication, there is a need for the authors to address the numbering discrepancy in reference [57], which is initially cited in Figure 1. Consequently, subsequent references should be adjusted.

Additionally, it is advisable for the authors to provide information regarding the gene fused to mCherry and to furnish details on how the resulting mCherry-fused protein facilitates the specific labeling of the node. This information could be helpful for the understanding of the process by non-specialized researchers.

Author Response

The order of reference citations follows the appearance in the main text first and then in the figure legends. However, we cited all references first in the main text in the revised manuscript.

In the revised manuscript, we explicitly stated that the node graft was from a transgenic quail expressing mCherry in all embryonic cells.

Reviewer 2 Report

Comments and Suggestions for Authors

While the authors address a potentially important topic in developmental biology, this review does not convey the importance of the topic, the relevance of the topic, the history of the topic, and the take home message.  A review should be written for interested scientists who may not read all the original literature in the field but want to get a clear overview.   This review does not do that.  There is "no forest -- it is all trees and weeds" -- to use the metaphor.   It is a dense compilation of thousands of  "facts" with  of reproductions from already published papers.   There is very little organization or a logical presentation of these "facts."   If one wanted this level of detail (with so little analysis or big picture overview) one would read all the original papers.   

If this is to be published the paper need to be completely rewritten.   It needs to begin (in the abstract and introduction) with a clear statement of the purpose of the review and why this is important for developmental biologists.   Then it needs to decide on a logic:  history of these cells (the review needs a clear delineation of the classic view and the new view  --WITH ---DIAGRAMS)   OR organization by topic.   The logic should be clear.  Then the authors should shorten this review by half and present schematic diagrams NOT reproductions from the literature.  The goal here is to integrate all the different literature .. and provide consensus and interpretation. 

------------

The abstract is poorly written; it does not convey the importance of these cells and why anyone should care.  Neither does it convey whether these cells are in all vertebrates or just some.  Neither does the abstract convey the goal of the review and what it will cover. 

Syntax:  Avoid one sentence paragraphs and really short paragraphs; this makes it very hchoppy and very hard to follow the logic of this review. 

l15: in abstract: "which are labeled by Sox2 N1 enhancer 15 activity" -- this needs explanation since it is not standard "gene expression";  if it promotes SOx 2 then why not Sox 2 expression?  This is answered later, but enhancer activity is an odd marker since it requires a transgenic approach. 

The abstract ends with:  Finally, the current status of in vitro models of the 24 NMP-dependent developmental process is reviewed.    However the word finally makes no sense because there is no indication of what will be reviewed before in vitro  models?    In Vivo models?   This is very unclear. 

The authors need to describe this enhancer activity marker in the introduction as well; this entire history is very sketchily done. They need to elaborate and explain  if they introduce the history of the discovery.  the history of the discovery is impossible to understand they way they have written it. 

l 158:  They write: In our quest to understand the mechanisms of Sox2 regulation in embryonic neural 158 tissue development, we have identified nearly 30 enhancers that activate Sox2 in the CNS, neural crest, and sensory placod. es in chicken embryos [16, 17, 18] in a developmental 160 stage- and embryonic domain-specific manner. --- Is this a review or a description of the authors work?   This is very unclear.

Basically the entire review is very unclear ... it is simply a compilation, a cut and paste from published literature that lacks overall analysis and insight.  The point of a review is to do the work and integration for the reader, to make sense of the field. This review does not succeed in that. At the end of the review one still does not know how broad this phenomenon is phylogenetically and how wide accepted it is by investigators or what the current view AND future research should be, much less the importance.  

Author Response

We virtually rewrote the manuscript in a logic-oriented fashion, substantially shortened the manuscript by avoiding references to the details of the original data, reduced the number of main figures from 13 to 9, and provided more schematic representations in the figures than in the previous version.

Round 2

Reviewer 2 Report

Comments and Suggestions for Authors

The current revised manuscript is much improved over the first version of this paper.  While I am willing to accept its publication, it still remains very dense and non-accessible to those not in this field.   It would still benefit from explaining to the general developmental biologist why this should matter.   It still has elements of esoteric anatomy descriptions and does not make connections to the broader field of developmental biology.  The paper would be much stronger if they made cogent arguments why understanding this tissue type is of more general importance.    However given that this may be useful to investigators working in this exact field, I am willing to leave this to the editor's discretion.  However this still feels like a lost opportunity to explain the importance and relevance to the broader field.